# Enhancing Swimming Performance by Optimizing Structure of Helical Swimmers

**DOI:** 10.3390/s21020494

**Published:** 2021-01-12

**Authors:** Jiaqi Miao, Xiaolong Li, Bo Liang, Jiongzhe Wang, Xiaofei Xu

**Affiliations:** R&D Institute of Fluid and Power Engineering, Dalian University of Technology, Dalian 116024, China; jackson.miao.527@gmail.com (J.M.); xiaolong__li@mail.dlut.edu.cn (X.L.); bo_liang@mail.dlut.edu.cn (B.L.); jiongzhe_wang@mail.dlut.edu.cn (J.W.)

**Keywords:** biomimetic robots, resistive force theory (RFT), structure optimization, flagellar motion

## Abstract

Untethered microrobots provide the prospect for performing minimally invasive surgery and targeted delivery of drugs in hard-to-reach areas of the human body. Recently, inspired by the way the prokaryotic flagella rotates to drive the body forward, numerous studies have been carried out to study the swimming properties of helical swimmers. In this study, the resistive force theory (RFT) was applied to analyze the influence of dimensional and kinematical parameters on the propulsion performance of conventional helical swimmers. The propulsion efficiency index was applied to quantitatively evaluate the swimming performance of helical swimmers. Quantitative analysis of the effect of different parameters on the propulsion performance was performed to optimize the design of structures. Then, RFT was modified to explore the tapered helical swimmers with the helix radius changing uniformly along the axis. Theoretical results show that the helical swimmer with a constant helix angle exhibits excellent propulsion performance. The evaluation index was found to increase with increased tapering, indicating that the tapered structures can produce more efficient motion. Additionally, the analysis method extended from RFT can be used to analyze the motion of special-shaped flagella in microorganisms.

## 1. Introduction

High-precision interventional small-scale continuum robots have shown great potential in biomedical applications [1]. In recent years, in order to achieve non-invasive surgery or targeted drug delivery in hard-to-reach areas of the human body, numerous untethered microrobots have been proposed to perform diverse tasks at the microscale level [2,3,4,5,6], ranging from diagnostic and therapeutic tasks in vivo to probing, analyzing, and transporting micro-objects in biology, to fluidic applications in lab-on-a-chip devices. However, since the flow around them is at a low Reynolds number, executing a geometrically reciprocal motion will lead to no net displacement according to the scallop theorem [7]. Thus, small scale organisms or artificial microrobots must perform complex non-reciprocal motions to realize propulsion at a low Reynolds number.

In recent years, researchers have adopted a variety of motion mechanisms based on bionics to drive microrobots and achieve efficient motion performance [8,9,10]. The propulsion mechanism of prokaryotic bacteria has attracted significant attention [11,12]. Due to the asymmetry of the helical structure of prokaryotic bacteria, rotating helical flagellum performs non-reciprocal motions which makes bacteria move at a low Reynolds number effectively. The action of rotating helical flagella is used to drive their cell body, and the bundling and unbundling of the flagella are controlled to change the direction of movement [13]. Understanding the forces acting on the flagellum via a kinematic model is important in designing the helical swimmer. Till now, many models, such as resistive force theory (RFT) [14], boundary element method (BEM) [15], and slender body theory (SBT) [16], have been proposed to obtain the hydrodynamic force acting on moving helical swimmers. SBT applies to flagellar structures whose helix radius is extremely small compared to the flagella length. Under such conditions, the calculation accuracy is higher. BEM approximates the solution to a partial differential equation (PDE) by looking at the solution to the PDE on the boundary and then uses that information to find the solution inside the domain. It can realize high-precision calculations of the motion of flagellum with different shapes [17]. However, its calculation process is extremely complicated. RFT depends on the choice of resistance coefficients, and it can fully meet the needs if the accuracy requirements are not very high.

Several studies have been carried out on the practical medical applications of helical swimmers. Examples include miniaturization of the helical swimmer [11], motion control and path planning [18,19], simulation of movement in blood-like flow [20], biocompatibility [3,10], and various driving methods [21,22,23,24,25]. However, the limitation of swimming velocity and insufficient efficiency are common problems in these studies. Even if helical swimmers travel at a velocity that is tens of body lengths per second, these problems still exist in actual medical tasks. Previous studies have shown that the dimensional parameters and rotating frequency of the helical swimmer have a great impact on the propulsion performance [26,27,28,29]. A dimensionless study on the geometrical parameters is also proposed by using Design of Experiments (DoE) [30]. Meanwhile, the helical swimmer driven by infinite power sources (e.g., magnetic field) has sufficient energy, which means that it does not need to consider propulsion efficiency. The ultimate goal is to obtain a structure that can generate the maximum thrust, even if it is not highly efficient. By contrast, the helical swimmer, driven by finite power sources (e.g., the chemical driving method [31] and biological driving method [32]), also needs to consider propulsion efficiency. The propulsion efficiency is given priority over propulsion capability (i.e., the thrust). In addition to using the optimal structure parameters and increasing the rotating frequency to improve the propulsion performance, the bundled flagella [33,34] and the introduction of hydrophobic materials [35,36] are also effective methods. Considering the fabrication and the control of microrobots, keeping the structure simple is the basic criterion.

In this paper, a mathematical model is established to analyze the swimming performance of helical swimmers. The propulsion efficiency evaluation index was introduced to evaluate the swimming performance of helical swimmers. Based on RFT, the influence of various parameters on propulsion performance is quantitatively analyzed, and the extent of the influence of each parameter on propulsion performance is given. Subsequently, to explore new methods to improve the propulsion performance of helical swimmers, two types of tapered helical swimmers are proposed. Based on previous theoretical results, RFT is modified to analyze the swimming performance of the tapered helical swimmers. In addition, the model for tapered structures is extended to the helical structures with a frustum of a cone shape, and more related helical structures are discussed. In summary, the swimming performance of helical swimmers is analyzed and discussed from two aspects (parameters adjustment and structure optimization), which provide a reference for the design of helical swimmers. The method for analyzing the propulsion ability of tapered helical swimmers can also be used to analyze some special-shaped microbial flagella.

## 2. Methods

### 2.1. Theoretical Analysis of Rigid Helical Swimmers

The Reynolds number plays an important role in characterizing the flow condition. The rotating Reynolds number of the rigid helical swimmers in the liquid phase can be defined as:(1)Re=ρωR2μ⇔inertial forcesviscous forces
where *ρ* and *μ* are the liquid density and dynamic viscosity, respectively, *ω* is the angular velocity of the flagellum, and *R* is the characteristic dimension of the helical swimmer. Due to the low Reynolds number of helical swimmers, the inertial force is considered negligible. The Navier-Stokes equation can be simplified as follows:(2)$$\nabla p=\mu \nabla^{2}\vec{U}$$
where *p* is the pressure scalar field, and $\vec{U}$ is the velocity vector field. This equation is independent of time and is linear. The propulsion motion along the axis of the helical swimmer is described by [7]:(3)$$\left[\begin{array}{l} F\\ T \end{array}\right]=[\begin{array}{l} B\\ C \end{array}\;\begin{array}{l} C\\ D \end{array}]\left[\begin{array}{l} u\\ \omega \end{array}\right]$$
where *B*, *C*, and *D* are the matrix coefficients, which are determined by dimensional parameters of the object and fluid physical properties. The equation shows that the relationships between the swimmer’s velocity *u*, angular velocity ω, external force *F*, and external torque *T* are related linearly. As for the motion of asymmetric structures like helical swimmers, *C* is not zero. This means that an external torque acting on the helical swimmers can drive a linear motion.

The resistive force theory (RFT) proposed by Gray and Hancock [14] can be used to calculate the thrust and the torque generated by the movement of the helical swimmer. Its basic assumption is that fluid resistance is proportional to the velocity. Based on this, the total thrust and torque can be obtained by integrating the force and the torque acting on every micro-element of the helical swimmer along the axial direction. Figure 1 shows all the parameters and the forces acting on the micro-element *ds* of helical structure (shown in front view, which will be used in this paper as the main view).

The thrust *dF_z_* and the torque *dT_z_* acting on the micro-element *ds* in the z-axis direction depend on the tangential force component *dF_t_* and the normal force component *dF_n_*.
(4)$$dF_{z}=dF_{t}\sin\theta -dF_{n}\cos\theta$$
(5)$$dT_{z}=-A(dF_{t}\cos\theta +dF_{n}\sin\theta )$$
where *θ* is the helix angle, and *A* is the helix radius. The normal force component *dF_n_* and the tangential force component *dF_t_* can be expressed as:(6)$$dF_{n}=C_{n}V_{n}ds$$
(7)$$dF_{t}=C_{t}V_{t}ds$$
where *C_n_* and *C_t_* are the drag coefficients in the normal direction and in the tangential direction, respectively. The normal velocity *V_n_* and the tangential velocity *V_t_* are respectively expressed as:(8)$$V_{n}=A\omega \sin\theta -v\cos\theta$$
(9)$$V_{t}=A\omega \cos\theta +v\sin\theta$$
where *v* is the velocity of the helical swimmer, and *ω* is the angular velocity. The drag coefficients in the normal direction and in the tangential direction can be given by [37]:(10)$$C_{n}=\frac{4\pi \mu }{\ln(2\lambda /d)+0.5}$$
(11)$$C_{t}=\frac{2\pi \mu }{\ln(2\lambda /d)-0.5}$$
where *µ* is the fluid viscosity, and *λ* and *d* are the pitch and the line radius of the helical swimmer, respectively. Based on the above equations, the thrust *F_z_* and the torque *T_z_* exerted on the helical swimmer can be expressed as:(12)$$F_{z}=\int_{z=0}^{z=l}dF_{z}=N\lambda \left[\left(C_{t}-C_{n}\right)A\omega \cos\theta +v\left(C_{t}\sin^{2}\theta +C_{n}\cos^{2}\theta \right)\csc\theta \right]$$
(13)$$T_{z}=\int_{z=0}^{z=l}dT_{z}=N\lambda A\left[\left(C_{t}-C_{n}\right)v\cos\theta +A\omega \left(C_{n}\sin^{2}\theta +C_{t}\cos^{2}\theta \right)\csc\theta \right]$$
where *l* is the axial length, and *N* is the number of helical turns.

### 2.2. Modified Resistive Force Theory (RFT)

Based on the mathematical model in this article, one side of the helical swimmer is fixed, and its swimming ability is determined by measuring thrust and torque. When analyzing helical swimmers in motion, the balance between fluid resistance and the swimmer’s propulsion need to be considered. However, in the present model of this paper, the helical swimmer is assumed to be bound in a fixed position when rotating, which leads to the need for corrections when calculating the theoretical values. The velocity item *v* in Equations (12) and (13) is zero. Therefore, the thrust and torque are expressed as:(14)$$F_{z}=N\lambda \left(C_{t}-C_{n}\right)A\omega \cos\theta$$
(15)$$T_{z}=N\lambda A^{2}\omega \left(C_{n}\sin^{2}\theta +C_{t}\cos^{2}\theta \right)\csc\theta$$

To provide a method to calculate the propulsion performance of the helical swimmers, RFT is modified for analysis. The helical swimmer is divided into countless micro-elements, which is the analysis method adopted by RFT. With the same idea, the tapered structure can be divided into *n* micro-elements (n approaches infinity). Each micro-element with a length of *dl* is regarded as the structure with a constant helix radius. The force *dF*_i_ and torque *dT*_i_ of each micro-element can be expressed as:(16)$$dF_{i}=dl\cdot F_{z}/l=dl\cdot \left(C_{t}-C_{n}\right)A\omega \cos\theta$$
(17)$$dT_{i}=dl\cdot T_{z}/l=dl\cdot A^{2}\omega \left(C_{n}\sin^{2}\theta +C_{t}\cos^{2}\theta \right)\csc\theta$$

The final results can be obtained by superimposing the micro-elements along the axial direction. Thus, if the thrust and the torque on them are expressed as *dF*_i1_, *dF*_i2_, …, *dF*_in_ and *dT*_i1_, *dT*_i2_, …, *dT*_in_, respectively, the total thrust and torque can be given by:(18)$$F=\sum_{k=1}^{n}dF_{ik}=dF_{i1}+dF_{i2}+\ldots +dF_{in}$$
(19)$$T=\sum_{k=1}^{n}dT_{ik}=dT_{i1}+dT_{i2}+\ldots +dT_{in}$$

### 2.3. Establishment of the Evaluation Index: Propulsion Efficiency Evaluation Index (K)

Under the driving conditions of limited energy supply (e.g., chemical driving and biological driving), the propulsion efficiency of helical swimmers need to be considered. Here, the evaluation index *K* is obtained via the same method as the previous work [26], which is defined as Equation (20):(20)$$K=\frac{F}{\mu A\lambda \omega }=\frac{F}{2\pi \mu A\lambda f}$$

## 3. Results and Discussion

### 3.1. Basic Model

Firstly, a basic model was established to analyze diverse parameters. A scale-up system was set up. It had the same low Reynolds number flow field as microrobots operated in biofluids. By reducing the rotating frequency and increasing the viscosity, the enlarged model could also achieve the result that the viscous force would be much larger than the inertial force. Scaled-up robot prototypes could help us design and control principles for microscale robotics systems. The actual measurement value in the experiment had a small error compared with the result obtained from the scale-up system, which would make this method widely adopted by researchers [26,38]. The related dimensional parameters are marked in Figure 2. The basic model had a helix angle of 45°, a helix radius of 5 mm, a line radius of 0.5 mm, a pitch of 3.14 cm, a rotating frequency of 3 Hz, and it was operated in the fluid with a kinematic viscosity of 30,000 cSt. The Reynolds number was about 10^−2^, which is in the low Reynolds number flow. Changing the parameters based on this model could control the Reynolds number to change in the range of 10^−3^–10^−1^, and then theoretical analysis could be made. Most microrobots for medical applications are in this range, thus, the analysis has great guiding significance for the design of helical swimmers.

### 3.2. Effect of Dimensional and Kinematical Parameters on the Propulsion Performance of the Basic Model

To investigate the specific impact of every parameter on the propulsion performance, the value of every single parameter was changed based on the basic model to obtain the corresponding relationship. The Reynolds number was controlled at the range of 10^−3^–10^−1^. The influence of the parameters (line radius, helix radius, pitch, length, and frequency) on the propulsion performance indexes (thrust, torque, and *K*) was examined (with the software MATLAB 2016b). The theoretical results are shown in Figure 3.

It can be seen that an increase in the line radius enhances *K*, which is consistent with the result of previous studies [26]. However, according to the assumption of the slender body, the line radius cannot be too large. Helix radius, pitch, and helix angle were mutually restricted, and satisfied the following relation:(21)$$\tan\theta =\frac{\lambda }{2\pi A}$$

The influence of the helix radius and pitch is shown in Figure 3b,c. The increase in axial length or rotating frequency did not bring about changes in *K*. In the structure design of helical swimmers, the appropriate size should be selected according to the application scenarios, and attention should be paid to avoid the clunky structure caused by excessive length. Although increasing the rotating frequency could not improve *K*, it could reach a higher thrust. Therefore, it was necessary to increase the rotating frequency as much as possible to improve the propulsion ability of a helical swimmer. It should be noted that there was a cutoff frequency under the condition of magnetic field driving [39]. Thus, the frequency needed to be controlled below it to obtain the optimal propulsion performance.

### 3.3. Quantitative Analysis of the Effect of Various Parameters on the Propulsion Performance

Under the influence of these five parameters, it was significant to grasp the main influence on the structure design of helical swimmers. Orthogonal analysis and range analysis were combined to evaluate the extent of influence of diverse parameters on the propulsion performance (which can be seen in Appendix A). The analysis results are shown in Figure 4 (Data analysis is done through the software Minitab 17).

The extent of influence of each parameter on a certain propulsion performance index depended on the range of k1 to k4. It can be seen in Figure 4 that the order of the extent of influence was as follows:Thrust: frequency > helix radius > length > pitch > line radiusTorque: pitch > helix radius > frequency > length > line radius*K*: pitch > frequency > line radius > helix radius > length

These results could provide strategies for a certain propulsion performance index to be improved, giving priority to changing the parameter that had the greatest impact on this index.

### 3.4. Analysis of the Propulsion Performance of Tapered Helical Swimmers

In addition to the optimization of size parameters, the exploration of structure optimization may also bring some interesting results. To further improve the propulsion performance of the helical swimmer, the strategy of structure optimization was adopted.

Based on the extension of RFT in Section 2.2, the tapered helical swimmer, with uniformly changing helix radius along the axial direction, was analyzed. For the convenience of comparing the propulsion performance, tapered helical swimmers were changed from the basic model. In Figure 5, it shows two types of tapered helical swimmers with uniformly changing helix radius along the axial direction. One keeps the pitch constant, and the other keeps the helix angle constant. Both of them had a maximum helix radius of 5 mm (*A*_n_ = 5 mm) and a minimum helix radius of 0 mm (*A*_1_ = 0 mm). As shown in Figure 5, the helix radius decreased uniformly along the axial direction. If the structure was divided into n micro-segments (n approaches infinity), then each micro-segment could be seen as the structure with an equal helix radius, which has been discussed before. Take the calculation in Figure 5a as an example. Firstly, thrust and torque produced by the structure with a constant pitch and a change of helix radius from 0 mm to 5 mm could be obtained. The points on the obtained curve represented a 9.42 cm long helical structure with the same pitch but a helix radius varying from 0 mm to 5 mm. After dividing them by the length of 9.42, they were just the micro-elements divided from the tapered helical swimmer with a constant pitch. Distribute them uniformly on the length of 9.42 and integrate, then the final thrust and torque of the tapered helical swimmer could be obtained. The structure in Figure 5b is analyzed in the same way. The only difference was that the analysis kept the same helix angle instead of the same pitch when the helix radius changed from 0 mm to 5 mm to get the corresponding curve. The calculation process of two tapered helical swimmers is shown in Figure 6b,c.

The three propulsion performance indexes are listed in Table 1. As for the calculation of *K*, the two types of tapered helical swimmers were different. For the tapered swimmer with a constant pitch, *λ* was constant (i.e., 3.14 cm) and *A* was (*A*_0_ + *A*_1_)/2 = 2.5 mm. As for the tapered swimmer with constant helix angle, *A* was (*A*_0_ + *A*_1_)/2 = 2.5 mm, and *λ* was (3.14 cm + 0 cm)/2 = 1.57 cm. Among them, *K* of the tapered helical swimmer with constant pitch decreased by 24.4%. By contrast, the tapered helical swimmer with a constant helix angle had a greater improvement. *K* had increased by 111.2%. It meant that they could perform better than the basic model under any driving method (including the ‘infinite energy’ field and the ‘limited energy’ field).

Next, was further discussion on the transitional structures from the basic model to the tapered structures, that is, the helical swimmers with a frustum of a cone shape (0 mm < *A*_1_ < 5 mm). It shows in Figure 7 that the evaluation index *K* of the tapered helical swimmer with a constant helix angle was improved with the reduction of *A*_1_. As for the tapered swimmer with a constant pitch, *K* decreased with the reduction of *A*_1_.

A similar tapered flagellated swimmer has also been proposed in a previous study [40]. By applying RFT and balancing the viscous drag with the elastic restoring force on the micro-element, they established the governing equation of the model. It was focused on the comparison of swimming performance between planar wave and helical wave. Additionally, the influence of taper ratio (*d*_f_/*d*_i_, where *d*_f_ is the diameter of the flagellum at the proximal end and *d*_i_ is the diameter of the flagellum at the base) on efficiency was analyzed. However, less analysis was discussed regarding different tapered structures. By contrast, the present study was focused on the tapered structure and the major aim was to provide a simple method to evaluate the corresponding propulsion performance of helical swimmers with different size parameters and structures. The analysis in this paper focused on the transition from the tapered swimmers to the helical swimmers with a constant helix radius. It was shown that with a decrease of the degree of tapering (i.e., *A*_0_ remained unchanged and *A*_1_ increased gradually as shown in Figure 7), the efficiency of tapered swimmers with a constant helix angle would increase, while the tapered swimmers with a constant pitch would decrease.

### 3.5. The Propulsion Performance of Special-Shaped Flagellum

This method extended from RFT could also be used to calculate the mechanics of parameters of some special-shaped helical structures. It was not only suitable for finding the optimal design for helical swimmers, but also for analyzing the propulsion performance of actual flagellum with a special shape. Here, the three possible types of flagella are listed in Figure 8 and their swimming abilities are analyzed in turn.

The first type is shown in Figure 8a. The flagellum is not an ideal structure with a strictly constant pitch. It may consist of multiple segments of flagella with different pitches. This situation is not uncommon. In nature, even the sperm heads of some animals are helical-shaped and can affect the movement [41]. The helical structure of each segment was regarded as a standard structure that could be obtained by RFT. The thrust and torque of each segment were *dF*_i1_, *dF*_i2_, …, *dF*_in_ and *dT*_i1_, *dT*_i2_, …, *dT*_in_, respectively, and the resultant force and torque were the sum of a set of finite discrete values. The second type is shown in Figure 8b. This flagellum may also show a gradual increase or decrease of the pitch. If this process was regarded as uniform, only the pitch was changing. According to the method proposed in this paper, if the flagellum was divided into countless micro-elements, then each micro-element could be regarded as a structure with a constant pitch. Based on the above method, the thrust and the torque generated by the flagellum could be obtained. After that, the propulsion performance of the flagellum could be evaluated. The third type of flagellum (such as a sea urchin sperm [42]) is shown in Figure 8c. It was the same as the tapered helical swimmers analyzed in this paper. The propulsion performance could be analyzed with the method in Section 3.4. The method proposed in this paper provides a strategy for the design of highly efficient helical swimmers. Under the assumption that the dimensional parameter changes uniformly along the axial direction, the propulsion performance of special-shaped flagella can be obtained. Not only can it be used to guide the design of helical swimmers applied in actual medical tasks, but it also provides an effective method for analyzing the movement of microorganisms with strange shapes.

## 4. Conclusions

In this paper, three indexes (thrust, torque and propulsion efficiency evaluation index *K*) have been applied to further improve the performance evaluation system of helical swimmers. They respectively provide evaluation criteria for selecting the optimal design for ‘infinite sources’ driving (magnetic field, light field, etc.) and ‘limited sources’ driving (chemical driving, micro-motor driving, etc.). First, the influence of five parameters (line radius, helix radius, pitch, axial length, and rotating frequency) on the three propulsion performance indexes (thrust, torque, and *K*) of helical swimmers is investigated. Then, orthogonal analysis and range analysis are used to quantitatively analyze the extent of influence of each parameter on the three propulsion performance indexes. In addition, more exploration is placed on optimizing the structure of helical swimmers. Two tapered helical swimmers are proposed, and RFT is extended to get a new method for analyzing the propulsion performance of them. The results show that the tapered helical structure with a constant helix angle is the best among all the structures. *K* increases during the gradual reduction of helix radius at the end of the helical swimmer (i.e., *A*_1_). It means that the new structure has better propulsion performance than the traditional helical structure under two types of driving conditions (‘infinite source’ and ‘finite source’). Meantime, it does not introduce a complex structure, which meets the actual medical needs of microrobots. This new structure has the potential to be used in low Reynolds number biofluids (cerebrospinal fluid, blood flow, urinary system, etc.) to perform medical tasks. In addition, the method proposed in this study can also be used to analyze the swimming ability of microorganisms (bacteria, sperms, etc.) with special-shaped flagella, which can help researchers better understand the movement of them.

In a future study, swimmers at a similar scale will be built to further demonstrate the theories proposed in this paper, and more discussions will be made on the swimming performance of helical swimmers. In addition, it will be used as a swimming robot in silt. The high viscosity of silt guarantees a low Reynolds number environment, and our analysis at this scale is completely suitable for it. The analysis in this paper also has great significance in the design of micro-scale helical swimmers. Moreover, the tapered helical swimmer, due to its unique structure (closed at one side), will bring potential applications such as drug loading in biomedical applications. Overall, the tapered helical swimmer, which retains the simplicity of the structure, has potential applications in micro-nano robotics and some special macroscopic flow fields.

## Figures and Tables

**Figure 1 sensors-21-00494-f001:**
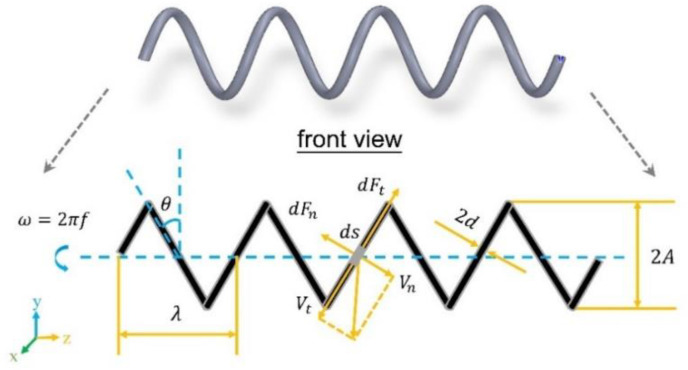
In the front view, the parameters of the helical swimmer can be expressed clearly, including *f* (rotating frequency), *θ* (helix angle), *λ* (pitch), *d* (line radius), *A* (helix radius), and forces acting on the micro-element *ds*.

**Figure 2 sensors-21-00494-f002:**
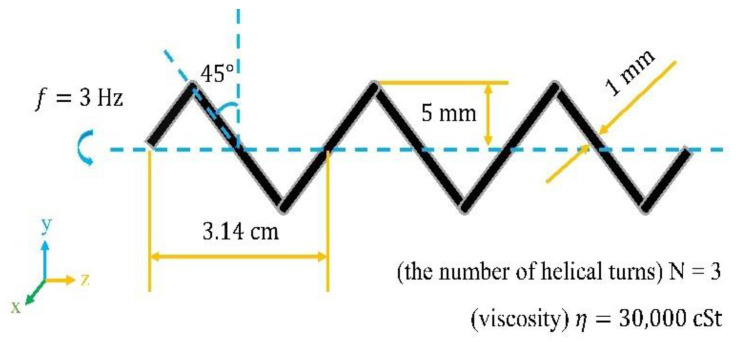
The related dimensional Basic model (*θ* = 45°, *A* = 5 mm, *d* = 0.5 mm, *λ* = 3.14 cm, *f* = 3 Hz, *N* = 3, and *η* = 30,000 cSt).

**Figure 3 sensors-21-00494-f003:**
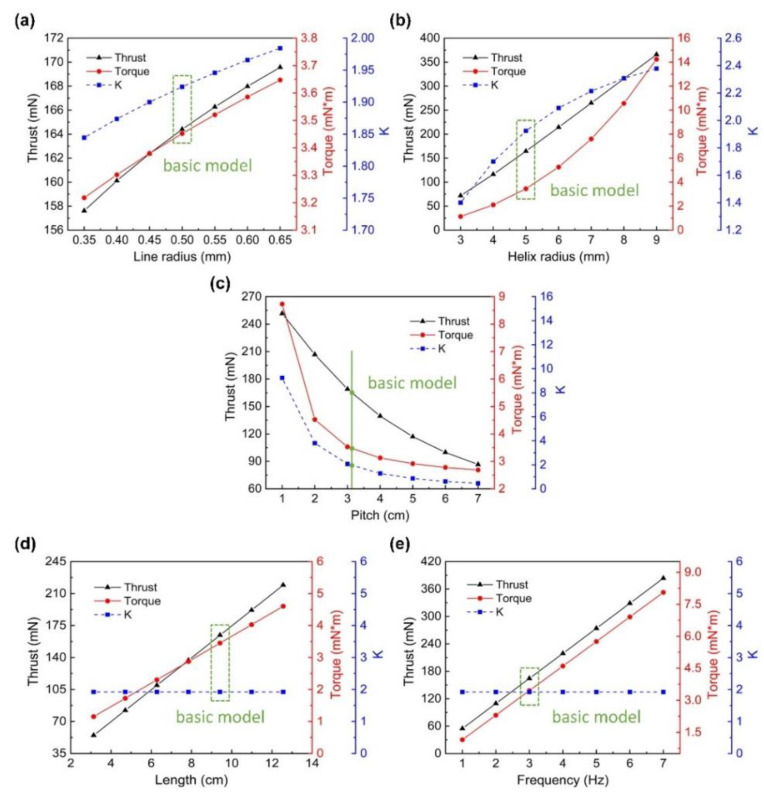
The influence of five parameters ((**a**) Line radius, (**b**) Helix radius, (**c**) Pitch, (**d**) Length, and (**e**) Frequency) on the three propulsion performance indexes (thrust-black line, torque-red line, and K-blue line). The locations of the basic model are marked in green.

**Figure 4 sensors-21-00494-f004:**
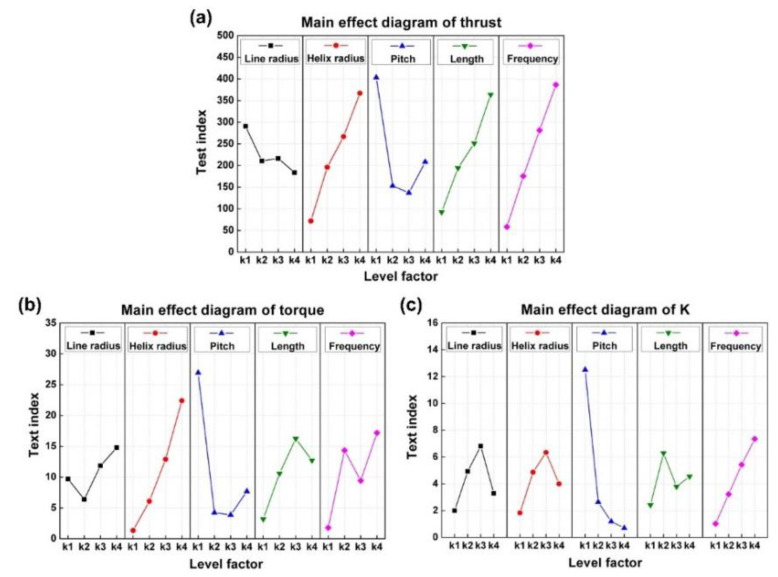
Main effect diagram of three propulsion performance indexes (The detailed data of Figure 4 are shown in Table A3, Table A4 and Table A5). (**a**) Thrust. (**b**) Torque. (**c**) *K*. Line radius, helix radius, pitch, length, and frequency are presented by the black line, red line, blue line, green line, and pink line, respectively.

**Figure 5 sensors-21-00494-f005:**
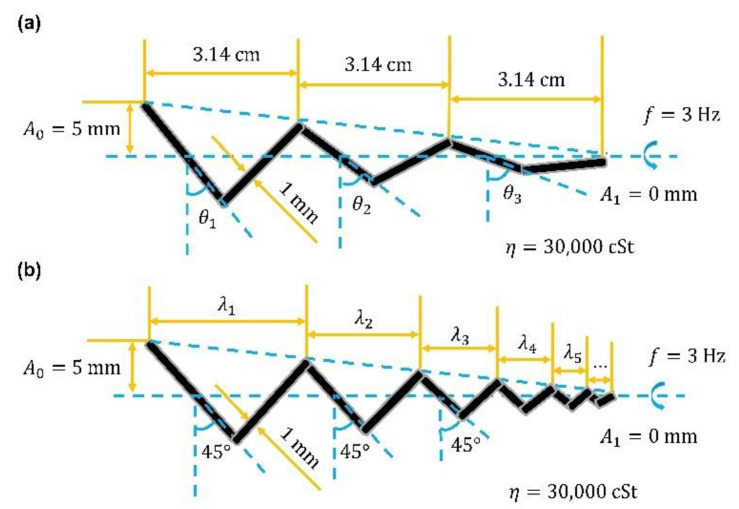
Two types of tapered helical swimmers. (**a**) The tapered helical swimmer with a constant pitch. (**b**) The tapered helical swimmer with a constant helix angle. Some main parameters are the same as the basic model (i.e., *d* = 0.5 mm, *λ* = 3.14 cm, *η* = 30,000 cSt, and *f* = 3 Hz).

**Figure 6 sensors-21-00494-f006:**
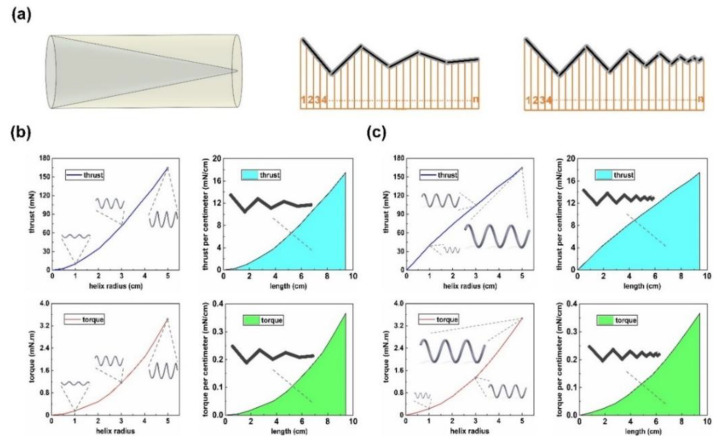
(**a**) On the left, it shows the enclosed volume of the two types of tapered helical swimmers (gray area) and the basic model (pale yellow area). On the right, it describes the calculation process of the two types of tapered helical swimmers. Both of them are divided into numerous micro-elements; The final thrust and torque are obtained by linear superposition of the micro-elements. The results of thrust and torque are in the blue and green area, respectively. (**b**) The tapered helical swimmer with a constant pitch is on the left; (**c**) The tapered helical swimmer with constant helix angle is on the right.

**Figure 7 sensors-21-00494-f007:**
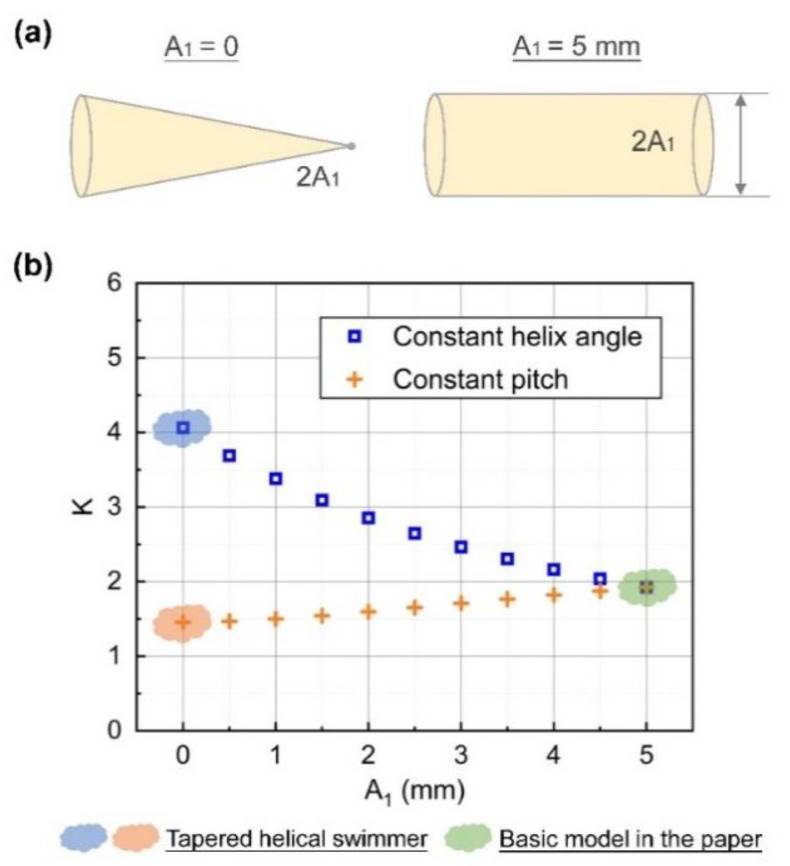
(**a**) When A1 (the minimum helix radius) change from 0 mm to 5 mm, the different enclosed volume is shown. (**b**) Changes in the evaluation index from the basic model to the tapered structures. When *A*_1_ = 0, it is the tapered structure. The blue mark represents the structure with a constant helix angle, the orange mark represents the structure with a constant pitch, and the orange mark represents the basic model.

**Figure 8 sensors-21-00494-f008:**
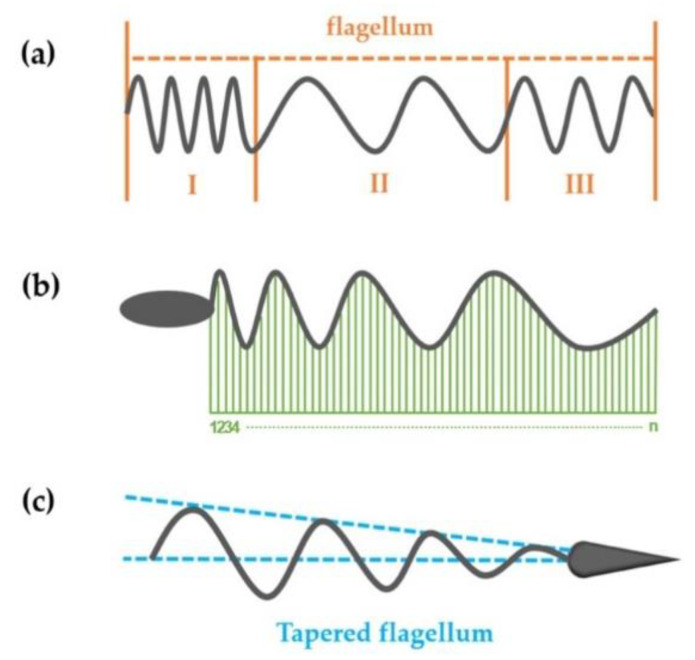
Three types of flagella. (**a**) The flagellum is composed of multiple segments of flagella with different pitches. (**b**) The flagellum with uniformly increased or decreased pitch. (**c**) The flagellum has a tapered helical structure like which has been discussed above.

**Table 1 sensors-21-00494-t001:** The three propulsion performance indexes (thrust, torque, and *K*) of three models (basic model, the tapered helical swimmer with a constant pitch, and the tapered helical swimmer with constant helix angle).

Model	Thrust (mN)	Torque (mN·m)	*K*
Basic model	164.40	3.45	1.924
Tapered helical swimmer with a constant pitch	62.15	1.10	1.455 (24.4%↓)
Tapered helical swimmer with constant helix angle	86.79	1.25	4.063 (111.2%↑)

## Data Availability

Derived data supporting the findings of this study are available from the corresponding author on request.

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
