# Peer review of "Enhancing Swimming Performance by Optimizing Structure of Helical Swimmers"

_sensors, 2021, doi:10.3390/s21020494_

Round 1

Reviewer 1 Report

I believe that the paper has improved a bit in its revised version. Some efforts have been made to make the methods clearer. However, some points in my initial review remain unaddressed, and I believe they are essential.

What the authors call 'modified RFT' (sect. 2.2) consists in numerically evaluating the known expressions given by RFT in eqs. (12) and (13). The tapering of the helix is taken into account by summing the results from these expressions for varying values of the parameters, such as the radius A. How these parameters are varied is not specified, as the equations used for the tapered helices are not given, though they can be guessed from Fig. 5. However, it seems to me that the integrals of eqs. (12) and (13) should lead to different results in the case where A or λ are function of z. For example, what is the value of N in the individual F_i? It is not defined locally. What would be correct instead would be to evaluate the integrals directly (either analytically or numerically).

I also still have some issues with the index sigma. I understand the motivations for choosing K, though such an index should be dimensionless. Concerning sigma, however, I found the justification quite insufficient. The author essentially just claim that the volume is 'the best' parameter without much arguments, besides the fact that the Navier-Stokes equations are usually written in force-density units. As I stated earlier, it is particularly critical considering that the volume is not kept constant between the different geometries. Furthermore, by contrast with the case of K, I see no reason why the usual index F/μAλω could not be used.

The third point concerns the Buckingham Pi theorem and the scaled-up system. I am still not sure why, or even if the Pi theorem has been used. The aim of the Pi theorem is to derive (sets of) dimensionless quantities from the variables at play. It seems that the authors have simply picked the parameters so that the scale is macroscopic but the Reynolds number small. I fail to see where the Pi theorem intervenes there.

Reviewer 2 Report

The reviewer appreciate the changes executed by the authors.

Author Response

We thank the referee very much for a recommendation of the publication in Sensors. We have improved some descriptions and revised English grammar. And we hope it could make you understand it clearly. The revised manuscript is in the attachment. Thanks again for your constructive comments.

Reviewer 3 Report

Comments and Suggestions for Authors

This manuscript presents a computational investigation of optimal geometric parameters for helical swimmers. Two indices for evaluating swimmer kinematic performance, thrust per unit volume and thrust per unit torque, are proposed. Resistive force theory is used to inform a modified mathematical model to predict swimming efficiency. The modified RFT model is also used to design/size helical swimmers with non-uniformly shaped flagella, such as those with varying pitch segments, which are to be corroborated by experiments in a future report.

The research design and numerical data are convincing, while the data and the depth of analysis and discussion are also generally appropriate. The overall the work is interesting and provides new insight into helical simmers, however there are a few major points that I suggest be addressed before recommendation for publication.

First, Resistive force theory (RFT) modeling of tapered flagellated swimmers has previously been explored by Kotesa, Rathore, and Sharma (DOI 10.1007/s12668-013-0105-6). This work should be cited and the results presented in this manuscript should be compared to the previous report.

While RFT is well established, it does have major limitations in comparison to BME, SBT, etc. There should be some discussion on the limitations of RFT how those limitations could affect the given results.

I am somewhat concerned to as to the implications of the modified RFT model to real systems, as in any real system internal forces would be stored within a partially fixed helix rotated at one end. The physical implications of the imposed zero velocity condition should be discussed.

‘Efficiency’ is mentioned described a number of times, in terms of motion efficiency and energy efficiency. It is suggested to explicitly define efficiency.

The final paragraph of the manuscript, while providing a preview of interesting future work in silt, does not appear to add much to the manuscript and could be edited/removed (what is ‘sports performance’?).

Equation 8 appears to have the incorrect sign (DOI:https://doi.org/10.1016/S0006-3495(02)75204-1)

In Figure 6, (c) is not labeled/described in the caption.

In Figure 7 (a) it is unclear what the geometric parameters used for A1=0mm to A1=5mm are. It is suggested to include this in the appendix.

Line 14: Why is the propulsion performance of current helical swimmers is still unsatisfactory?

Line 22: ‘50%. It means’ should be changed to the ‘50%, indicating’

Line 22: Line 22 stats that ‘Figure 3 show that no single parameter can simultaneously increase σ and K’, however from Fig. 3 b2 it appears that decreasing helix radius can be used to increases both σ and K.

Line29 “in vivo” → “in vivo,”

Line37 “more” → “significant”

Line40 “binding and the dispersal” → “bundling and unbundling”

Line41 “via the kinematic” → “via a kinematic”

Line41 “the moving helical swimmer” → “moving helical swimmers”

Line50 “Previous studies have shown that the dimensional parameters and rotating frequency of the helical swimmer have a great impact on the propulsion performance [24, 25]” → many more recent papers have explored this and should be cited.

Line50 “However, few studies consider the enclosed volume of the helical swimmer and the thrust simultaneously. Also, there is a lack of quantitative analysis of the impact of various parameters on the propulsion performance.” → This is incorrect, numerous papers have explored this in depth and should be cited.

Line58-59 “often requires higher propulsion efficiency” → This is incorrect

Line64 “hydrophobic materials [30]” → ‘A swarm of slippery micropropellers penetrates the vitreous body of the eye’ by Wu et. al.  (DOI: 10.1126/sciadv.aat4388) should also be cited.

Line66-67 “they are not suitable in actual medical applications.” → This statement need to be explained and justified.

Line74“to solve the limitation of parameter optimization” → what is the ‘limitation of parameter optimization’?

Line74 “Besides, the tapered structure” → “In addition, the model for tapered structures”

Line91 “time and linear” → “time and is linear”

Line93 “It shows” → “This equation shows”

Line295 “respectively. And” → “respectively, and”

Line348 “flagella, which helps” → “flagella, which can help”

Line350 “In the future study” → “In a future study”

In general the manuscript should be proof read to improve grammar and coherence.

Round 2

Reviewer 1 Report

In the latest version of the manuscript, the author have added some more context and references, as well as well as language corrections which I believe have improved the paper. Changing the definition of K to match the dimensionless definition commonly found in the literature is also an improvement. I still have a few concerns detailed below.

Concerning section 2.2, the authors have explained in their response how they proceeded to calculate F and T, but it does not correspond to what is written in this section of the paper. If I understand correctly, the authors calculate F and T for several helices of constant radius, then calculate their sum weighed by dl/l where dl is the length of each infinitesimal element of a given radius. If this is indeed what the authors have done, then expressions (14) through (17) should reflect that. As they are written now, equations (16) and (17) express the sum of the F and T of n helices, not the F and T of a single tapered helix. 

Of course, numerically evaluating the integrals (12) and (13) with some linear function A(z) and λ(z) in dF and dT would be the correct way. Whether the authors' method works or not is debatable, but they should at least state their hypothesis clearly in the manuscript.

Concerning index σ (section 2.3.1), I still find the justification for using the volume quite lacking. The fact that the tapered and non-tapered helices studied have not, as far as I know, been compared in a situation where their volumes are the same means that this index is simply not very useful.

As a side note, in line 22 of the abstract, part of a sentence seems to be missing (“indicating” ...).

Reviewer 3 Report

The authors have improved the manuscript in the revised submission, however there are still a few points that I suggest be addressed before recommendation for publication.

While DOI 10.1007/s12668-013-0105-6 is now cited, connections between the effects between the degree of tapering and efficiency should be compared.

Line 21-22: “the two evaluation indexes evidently increase, and especially the efficiency evaluation index doubles, indicating. It means that the tapered structure can realize efficient motion” → “the two evaluation indexes were found to increase with increased tapering, indicating that the tapered structures can produce more efficient motion”

Line 45: “wire radius” → “helical radius”

Line 45: “BEM is only related to the boundary, which applies to flagella bacteria of any shape, and the calculation is the most accurate.” → should be re-worded and cited.

Line66-67 “they are not suitable in actual medical applications.” → it is suggested that this line is further edited or removed as the provided justification only provides current challenges in fabrication and control and does not address why medical applicants are not achievable.

Line 289: “in previous” → “in a previous”

Line 355: “robots” → “robotics”

Again, the manuscript should be proof read to improve grammar and coherence.

Author Response

This manuscript is a resubmission of an earlier submission. The following is a list of the peer review reports and author responses from that submission.

Round 1

Reviewer 1 Report

In this paper, the authors present their analysis of helical microswimmers. They discuss how the thrust and efficiency of these swimmers are affected by various parameters. In particular, they compare two types of tapered helices with the regular helix commonly found in the literature. I have some serious concerns with this manuscript, which means I cannot recommend it for publication in its present form. In particular, I find that the main claim of the paper is not very well supported and the presentation of the methods used is not sufficient. More details can be found below.

The authors use Resistive Field Theory (RFT) as introduced by Gray and Hancock (Ref. 13). Equations (1) through (15) are a reminder of the theory. Therefore, eqs. (16) and (17) are all we have about the way the authors calculated their torque T and thrust F in the tapered case. Not only is this insufficient, it is also presented as a new approach despite being strictly equivalent to the definitions given earlier. Indeed, eqs. (16) and (17) propose to evaluate F and T as a sum of infinitesimal portions of the flagellum, divided along the z-axis, which is rigorously identical to the integrals given in eqs. (12) and (13). Furthermore, the terms in the sum are never clearly defined (the F_i and T_i, which should incorporate the dependency in z) and how this expression is evaluated (numerically or else) is not known. In the annex of their paper (Ref. 13), Gray and Hancock explicitly calculate the case where the amplitude changes along the flagellum, which is essentially the planar equivalent of the tapered helix presented here. They show (analytically) that only the average amplitude of the wave matters. Of course the authors may have reasons to follow a different approach, which I would like to know, but at the very least they should make their own method transparent.

Considering that the analytical result from Ref. 13 in the planar case differs from what the authors see here, it would be interesting to contrast these two situations. Unfortunately, such a comparison is made impossible by the evaluation indexes used by the authors. Why calculate thrust per unit volume? I understand that the thrust must be rescaled to not depend so much on the size of the swimmer, but why the volume and not, for example, a length or area? Or simply a force? Contrary to what the authors claim, previous studies usually rescale thrust and speed. From eq. (12), it is pretty clear that µAλω would be a natural choice to make thrust dimensionless, and it is in fact often rescaled like this in the literature. Using the average value of A for the tapered helix would also make a comparison with the planar case possible. In Ref. 13, what is evaluated is the speed divided by the frequency and the wavelength. The fact that the rescaling by the volume is not properly justified is especially concerning when one considers that the tapered helices have a volume 3 times smaller. Why not keep a constant volume between the different helices? If the thrust does not scale like the volume (and it seems that it does not), this would artificially make the tapered case look more effective. To make any claim of improvement over the regular helix convincing, this rescaling by the volume has to be rigorously justified. It is also unclear why the authors compare force and torque, which has the units of one over a length, instead of a more traditional measure of efficiency.

The rest of the paper is not much clearer regarding the methods used. For example, there are several mentions of the Buckingham PI theorem, but it is never clear how or why it is used. From what I understand, the authors justify working on a scaled-up system by the fact that the Reynolds number is the same. In section 3.1, the authors say that their scaled-up system is based on the PI theorem, but they also talk about experimental measurements, which is very confusing considering that no experimental method is presented. How exactly is the data of Fig. 3 produced? And if there is now experiment, then what is the motivation for working on a scaled-up system? Concerning the rest of the paper, particularly Figs. 3 and 4 as well as Appendix A, I am not convinced that such a brute force analysis of all parameters is necessary (or very helpful) considering that analytical calculations are possible, and in fact already exist in the non-tapered case. Of course, properly justifying the use of the proposed indexes is necessary for this analysis to carry any weight.

Finally, I found that the language could be improved. Excluding typos and other small errors, a few sentences are difficult to understand. For instance line 12 in the abstract (maybe ‘rotates’ is meant in lieu of ‘rotation’), or the beginning of the third paragraph (‘studies (...) have triggered concerns.’ and ‘however’).

Reviewer 2 Report

General comments

This paper presents an in-depth study on the artificial helical swimmers subject. The paper is focused on applying a modification of the Resistive Force Theory (RFT) in conventional helical swimmers to achieve a better propulsion performance. Also, two evaluation indexes were developed by the authors so that the propulsion performance can be quantitatively evaluated. Experimental results are presented to validate the proposed theory.

Comments and suggestions for Authors

1.- Based on the literature review, the topic of this paper is clearly of scientific soundness. Because of this, the authors must highlight even more their contribution; since only 5 lines (see lines 67 to 71) describe their achievements. It would be convenient to add some items with the aim of highlighting your contribution.

2.- What properties does the matrix equation (3) have? Is some kind of mapping from one space to another? and What happened if the inverse relation does not exist? Please explain such a mapping.

3.- The indexes proposed in section 2.3, Are always applicable? or when they are valid? Because fluids are very erratic in their movement and the forces originated by them tend to change very quickly.

4.- Based on the testbed you used for your validation, does the results you find will always keep despite the dimensions of a real helical swimmer? Because the indexes you proposed could lead to some percentage of error due to the “extraordinary” size of the testbed. Please clarify.

5.- Authors should remind that it is a good practice to give the corresponding mention to all that we use with the aim of showing either simulation or experimental results. In this regard, which software did the authors use to find the numerical results depicted in Figures of the section 3.3?

6.- What kind of perturbations can be act over the system? Is there some “special” perturbations to be considered in this kind of systems? If so, it would be nice if authors include some kind of table where the potential perturbations were specified.

7.- More experiments need to be added, since authors claim that the proposed indexes "improve the performance evaluation system of helical swimmers". Thus, some comparison should be done regarding the published literature.

8.- Authors should remember that the future work gives a clear idea of the impact of their research. Thus, it would be nice if they give some ideas of the future of this work.

8.- The proposed paper could be published in Sensors Journal after minor changes be executed. It is important to highlight that, nowadays, Sensors Journal is Q1 and the quality of the published works must be superior.